# Monitoring Disease Progression with Stable Diffusion: A Visual Approach

**Sourav Kumar**[1,2], **Shell Xu Hu**[3] , **Aaron S Coyner**[4], **Wei-Chun Lin**[4], **Susan Ostmo**[4], **Deniz Erdogmus**[5], **RV Paul Chan**[6], **Michael F. Chiang**[7,8], **J.Peter Campbell**[4], **Jayashree Kalpahty-cramer**[9], **Timothy Hospedales**[3,10] and **Praveer Singh**[9,11]

[1]*Radiology, MGH/Harvard Medical School, Charlestown MA* [2]*Electrical And Computer Engineering, Brown University, Providence, RI* [3]*Samsun AI Research, Cambridge, UK* [4]*Ophthalmology, Oregon Health & Science University, Portland, OR* [5]*Electrical and Computer Engineering, NorthEastern University, Boston, MA* [6]*Ophthalmology and Visual Sciences, University of Illionois at Chicago, Chicago, IL* [7]*National Eye Institute, National Institutes of Health, Bethseda, MD* [8]*National Library of Medicine, National Institutes of Health, Bethseda, MD,* [9]*Ophthalmology, University of Colorado School of Medicine, Aurora, CO* [10]*School of Informatics, University of Edinburg, Cambridge, UK* [11]*Moorfields Eye Hospital NHS Foundation Trust, London, UK*

**Editors:** Accepted for publication at MIDL 2024

## Abstract

Monitoring disease progression is quintessential for effective healthcare by providing early diagnosis, timely interventions and improved patient outcomes. In the recent years, diffusion models have become extremely popular for synthetic medical image generation. In this work, we combine Stable Diffusion with image editing through textual guidance and cross-attention reweighing to predict disease progression for patients enrolled for a Retinopathy of Prematurity (ROP) disease screening program. Our technique provides effective visualization for monitoring disease severity over time on new subjects.

**Keywords:** Diffusion Models, modelling disease severity, medical image synthesis.

## 1. Introduction

Retinopathy of Prematurity (ROP) is the leading cause of childhood blindness in premature infants, which is caused by the growth of abnormal vasculature in retina. While only 5-10% of total cases progress to severe ROP, it is quite important to understand the physiology as well as disease progression for early diagnosis and effective and timely treatment of ROP.

Recent works (Chen et al., 2021; Thambawita et al., 2022; Coyner et al., 2022; Han et al., 2018) have shown the efficacy of generative models for synthetic medical image generation, either for data augmentation or for privacy-preserved training of AI models. However, to the best of our knowledge, none of the works showcase longitudinal progression in imaging for the same patient via synthetic generation. Stable diffusion models have recently garnered significant attention in medicine, given their effectiveness in generating high-quality synthetic medical imaging(Chambon et al., 2022; Kim and Ye, 2022).

In this study, we work towards developing a tool to visualize the progression of ROP disease even in normal subjects, to better understand the physiology of the disease as well as to distinguish progression of normal eyes from diseased eyes. We leverage the Stable Diffusion model (Rombach et al., 2022) to build a class-conditioned ROP image generation

tool, which further enables a free transformation of disease severity for a given ROP image by combining recent image editing techniques. With our contribution, we hope to pave the way for more accurate and efficient ways for diagnosing as well as early intervention and timely treatment of this condition.

## 2. Methods

We cast the problem of visualizing ROP progression as an image editing problem. Given an input image of any of the Normal/Pre-Plus/Plus class, the goal is to transform it into another class, while preserving the global structure & some local attributes so that the transformed image is still perceived as an image belonging to the same patient. We first fine-tune a Text-to-Image SD model on the iROP dataset (5500 images), thus allowing the model to generate realistic images of each class prompted by textual inputs. We next employ two image editing techniques: Null Text Inversion (Mokady et al., 2022) & Prompt-to-Prompt Image Editing (Hertz et al., 2022), essentially allowing image modification through attention swapping and reweighting, Fig 1 during the forward diffusion process. To evaluate SD-based image generation, we calculate Frechet Inception Distance (FID) score between real & generated images as well as Plus disease classifier accuracy when tested on class-conditioned generated images. To evaluate image editing, we calculate the correlation between ROP Vascular Severity Score (VSS) of a generated image & real image of increased severity for the same normal image.

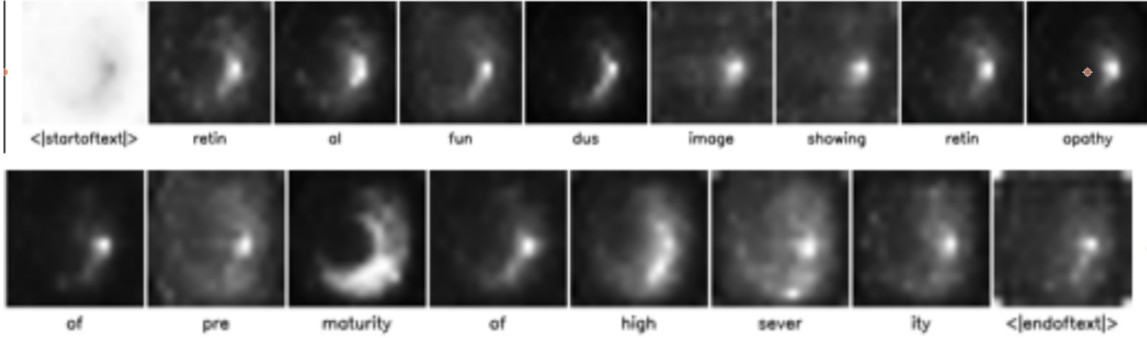

Figure 1: Prompt Attention Map showing image regions impacted by each word

## 3. Results

Our SD model achieves FID scores of 92, 114, and 137 for Normal, Pre-Plus, & Plus classes, respectively. We achieve 97% accuracy for generated plus class when tested with a Plus Classifier model. A correlation of 0.97 between ROP VSS of generated & real images indicates proportional attention weight adjustment enables the generation of required severity images. The predicted diseased samples of the real normal retinal images follow the same vascular structure, including dilation and tortuosity of blood vessels as they do in the natural progression of the disease, Fig 2, thus proving that this methodology can very closely predict the disease progression in any patient. Using this we can visualize the severity of

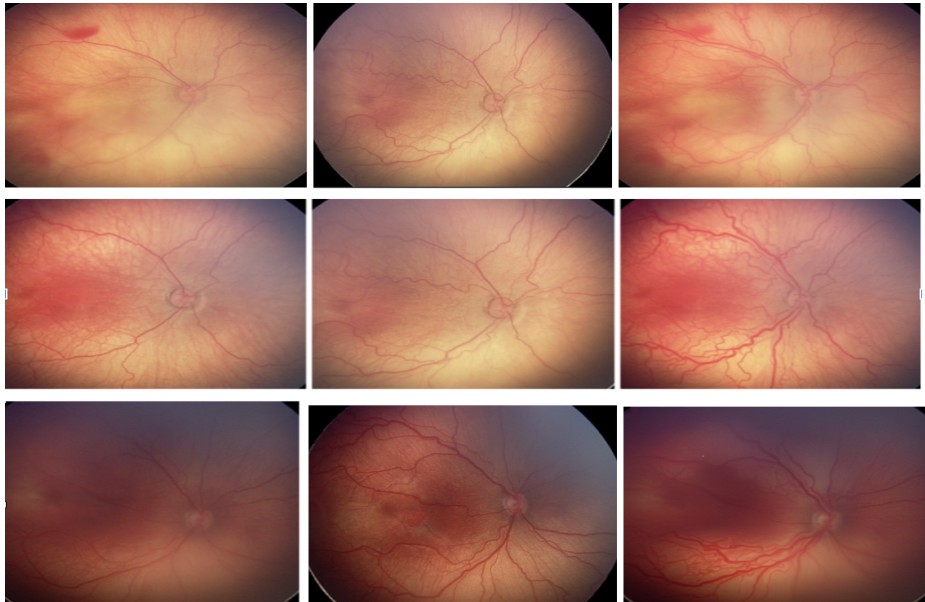

Figure 2: Actual (middle) vs Predicted disease progression (rightmost) of a normal retinal image(leftmost)

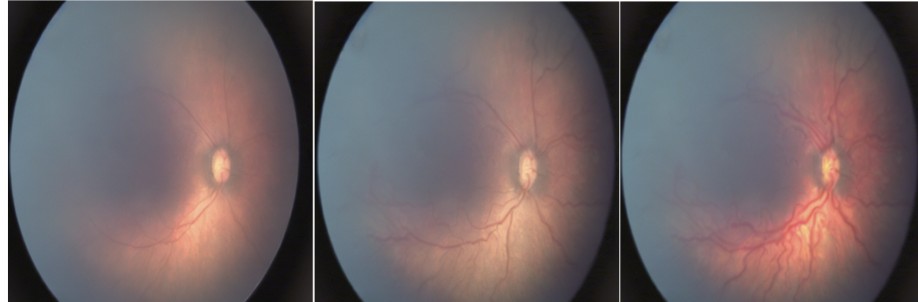

Figure 3: Predicting disease progression on a new sample with no progression yet. From left (Normal) to Pre-Plus(moderate severity) to Plus(Extreme severity of ROP- (right).

the disease at a future date on new data samples which have no disease development yet or are in the moderate disease severity. Fig 3.

## 4. Conclusion

Our method allows for visualization of the severity of the disease at a future date on new data samples, which may have no disease development yet or are at a moderate severity stage of the disease. This can provide clinicians with valuable insights into the potential progression of the disease and aid in early diagnosis and treatment, ultimately improving outcomes for premature infants.

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
