# OpenReview forum: "Monitoring Disease Progression with Stable Diffusion- A Visual Approach"
_MIDL.io/2024/Short_Papers — MIDL 2024 Short Papers_

### Official Review · Reviewer_fYBy · 2024-04-17

**Confidence:** 5
**Final Rating:** 5

**Review:**

This study explores the use of Stable Diffusion with image editing guided by text and cross-attention reweighting to forecast disease progression in patients participating in a Retinopathy of Prematurity (ROP) screening program. While the language is clear and detailed, and the application of SD for disease progression analysis is noteworthy, the technical innovation is modest as the work merges concepts from existing research. Nonetheless, the main contribution lies in the novel application of these methods for disease prediction.

---

### Decision · Program_Chairs · 2024-04-26

Accept